# TNFα Effects on Adipocytes Are Influenced by the Presence of Lysine Methyltransferases, G9a (EHMT2) and GLP (EHMT1)

**DOI:** 10.3390/biology12050674

**Published:** 2023-04-30

**Authors:** Ashley A. Able, Allison J. Richard, Jacqueline M. Stephens

**Affiliations:** 1Adipocyte Biology Laboratory, Pennington Biomedical Research Center, Baton Rouge, LA 70808, USA; 2Department of Biological Sciences, Louisiana State University, Baton Rouge, LA 70803, USA

**Keywords:** adipocyte, G9a, GLP, TNFα, methyltransferase

## Abstract

**Simple Summary:**

The enzymes, G9a and G9a-like protein (GLP), add methyl groups to protein targets. They can regulate gene transcription by modifying the proteins that organize DNA within cells, but they can also modify other targets independent of their enzyme activity. Studies show that G9a and GLP are important proteins in the development of metabolic disease, but their role in fat cell function is not well understood. Fat cells, known as adipocytes, help to regulate whole-body metabolic health by safely storing fat and by secreting molecules that communicate with other tissues, including the brain. In this study, we decreased the levels of G9a, GLP, or both enzymes in fat cells and examined the effects on adipocyte gene expression and fat breakdown, known as lipolysis, when fat cells were exposed to a proinflammatory protein, TNFα, that is associated with metabolic diseases states, such as Type 2 diabetes. The loss of both G9a and GLP enhanced TNFα’s ability to induce lipolysis and regulate gene expression. Our data also show that these enzymes form a complex with NF-κB, which is a signaling molecule induced by TNFα. These novel observations provide specific details on how G9a and GLP expression in adipocytes is associated with systemic metabolic health.

**Abstract:**

Impaired adipocyte function contributes to systemic metabolic dysregulation, and altered fat mass or function increases the risk of Type 2 diabetes. EHMTs 1 and 2 (euchromatic histone lysine methyltransferases 1 and 2), also known as the G9a-like protein (GLP) and G9a, respectively, catalyze the mono- and di-methylation of histone 3 lysine 9 (H3K9) and also methylate nonhistone substrates; in addition, they can act as transcriptional coactivators independent of their methyltransferase activity. These enzymes are known to contribute to adipocyte development and function, and in vivo data indicate a role for G9a and GLP in metabolic disease states; however, the mechanisms involved in the cell-autonomous functions of G9a and GLP in adipocytes are largely unknown. Tumor necrosis factor alpha (TNFα) is a proinflammatory cytokine typically induced in adipose tissue in conditions of insulin resistance and Type 2 diabetes. Using an siRNA approach, we have determined that the loss of G9a and GLP enhances TNFα-induced lipolysis and inflammatory gene expression in adipocytes. Furthermore, we show that G9a and GLP are present in a protein complex with nuclear factor kappa B (NF-κB) in TNFα-treated adipocytes. These novel observations provide mechanistic insights into the association between adipocyte G9a and GLP expression and systemic metabolic health.

## 1. Introduction

G9a (also known as euchromatic histone methyltransferase 2; EHMT2) and GLP (also known as euchromatic histone methyltransferase 1; EHMT1) primarily function to catalyze mono- and di-methylation of lysine 9 of histone H3 (H3K9) using the cofactor S-adenosyl methionine (SAM). Classically, mono-methylation of H3K9 is associated with transcriptional activation and di-methylation at this site is associated with transcriptional repression. Initially, studies indicated that G9a and GLP had to form a heterodimeric complex to exert their catalytic activity [1]. However, recent studies indicate that G9a and GLP have independent physiological functions [2,3], can methylate nonhistone substrates [4,5], can function as coactivators for a variety of nuclear receptors/transcription factors independent of their methyltransferase activity [3,6,7,8,9], and can act as transcriptional repressors [10,11]. Moreover, both G9a and GLP are capable of catalyzing the methylation of histone H3 lysine 23 [12], a discovery that adds to the complexity of these epigenetic enzymes.

Several studies have assessed the contribution of G9a and GLP expression in relation to metabolic health. G9a expression is decreased in the liver of db/db mice [3], an obese and diabetic mouse model that lacks a functional leptin receptor. Notably, when G9a is overexpressed in the db/db mice, hepatic insulin signaling is restored [3]. Additional in vitro experiments in HepG2 cells have shown that the loss of G9a protein levels results in the decreased expression of the insulin receptor alpha (IRα) and phosphorylated AKT and that increased G9a expression could prevent fatty acid induced insulin resistance [3]. However, experiments using a G9a/GLP-specific inhibitor that suppresses its catalytic function, BIX-01294, had no effect on IRα and phosphorylated AKT levels in HepG2 cells [3]. Together, these studies suggest that G9a is capable of modulating hepatic insulin signaling independent of its methyltransferase activity.

The cell-autonomous function(s) and underlying molecular mechanisms of G9a and GLP action have not been widely investigated in mature adipocytes. Mice lacking G9a in their adipocytes have increased adiposity, increased lipid accumulation in the liver, hyperinsulinemia, and decreased adiponectin in circulation [13]. Experiments using primary brown preadipocytes from G9a-floxed mice demonstrated that G9a suppresses adipocyte development through the repression of the PPARγ gene expression [13]. Additional studies have shown that G9a and the histone methyltransferase, SUV39H1, can also be recruited to the C/EBPα promoter to repress adipogenesis [14,15]. Other studies revealed that adipocyte-specific GLP-knockout mice have a reduction in brown adipose tissue (BAT)-mediated thermogenesis, increased adiposity, decreased insulin sensitivity, increased circulating insulin, and increased liver triglycerides [16]. Overall, these studies suggest that deletion of either G9a or GLP expression in adipocytes promotes a metabolically unfavorable phenotype in mice. Epigenetic modifiers, including methyltransferases, impact metabolic disease states such as obesity and Type 2 diabetes [17]. Epigenetic regulation not only contributes to disease development but is also relevant in responses to disease. Hence, understanding the role of methyltransferases in adipocyte function is merited.

Since the regulation and function of G9a and GLP in white adipocytes remains largely unknown, we investigated how G9a and GLP impact adipocyte function and gene expression in 3T3-L1 adipocytes. The 3T3-L1 adipocyte cell line is widely used to examine the basic cellular mechanisms associated with obesity and diabetes. Since in vivo studies indicate that G9a and GLP expression in adipocytes contributes to metabolic health in mice, we used an in vitro adipocyte model to determine if the loss of G9a and GLP expression could modulate the effect of the tumor necrosis factor alpha (TNFα) action in adipocytes. TNFα is a proinflammatory cytokine produced and upregulated in adipose tissue macrophages in conditions of obesity and Type 2 diabetes (T2D) [18], where it promotes lipolysis and induces insulin resistance. The adipose tissue TNFα acts in a paracrine manner on adipocytes and contributes to the development of insulin resistance through the induction of lipolysis, impairment in insulin signaling, and alterations in gene and protein expression [19]. TNFα promotes the expression and secretion of other proinflammatory cytokines, such as interleukin-6 (IL-6) [20]. TNFα also induces the expression of lipocalin-2 (LCN2) and the monocyte chemoattractant protein-1 (MCP1), a chemokine that contributes to macrophage infiltration into adipose tissue. Both LCN2 and MCP1 are upregulated in obesity and T2D in mice and humans [21,22]. Additionally, TNFα reduces the expression and secretion of the fat-specific hormone, adiponectin, and the insulin-regulated glucose transporter, GLUT4, which are both associated with insulin sensitivity [23,24,25]. 

Our studies are the first to demonstrate that the loss of either G9a or GLP in mature adipocytes does not impact mono- or di-methylation of H3K9 and does not impact basal or TNFα-induced lipolysis. In addition, these two methyltransferases are present in a complex in the nucleus of adipocytes. Moreover, we observed that the loss of both G9a and GLP in adipocytes did not have a significant effect on the mono- or di-methylation of H3K9 suggesting that G9a and GLP are not the primary H3K9 methyltransferases in adipocytes under basal conditions. However, when G9a and GLP levels were depleted in TNFα-treated adipocytes, we observed a significant decrease in the mono- and di-methylation of H3K9. Knockdown approaches revealed that the loss of both G9a and GLP in adipocytes also resulted in an enhancement of TNFα-induced lipolysis and proinflammatory gene expression. Coimmunoprecipitation experiments revealed that p65, the major functional subunit of NF-κB, is only present in a complex with G9a and GLP under TNFα-stimulated conditions in adipocytes. Collectively, these novel studies provide insight into the cell-autonomous roles of G9a and GLP in adipocyte gene expression and function that contribute to metabolic disease states.

## 2. Materials and Methods

The materials and methods associated with this study were previously published in Dr. Able’s doctoral dissertation [26].

### 2.1. Cell Culture

Murine 3T3-L1 preadipocytes (obtained from Dr. Howard Green at Harvard University) were grown to confluence and differentiated into lipid-laden mature adipocytes as indicated in [27]. The development of lipid droplets following differentiation was used to consistently confirm the identity of the 3T3-L1 adipocyte cell line. Prior to treatment with murine TNFα, the adipocytes were serum-starved overnight up to 24 h by changing the 10% serum medium to Dulbecco’s Modified Eagle’s Media (DMEM) (Sigma–Aldrich, St. Louis, MO, USA) containing 1% calf serum. Recombinant murine TNFα was purchased from Thermo Fisher (Waltham, MA, USA; Cat #: PMC3013). All cell culture experiments were performed in Corning^®^ Costar^®^ tissue culture (TC)-treated 10 cm or multiwell plates.

### 2.2. Small Interfering RNA (siRNA)-Mediated Knockdown

The knockdown of G9a or GLP individually or simultaneously using siRNA was performed in 6-well or 12-well plates as previously described [28]. Briefly, siRNA targeting G9a (Cat #: L-053728-01-0005) or GLP (Cat #: L-059041-01-0005) was purchased from Dharmacon (Lafayette, CO, USA). Media changes were performed every 24 h, and nontargeting siRNA acted as a negative control for the knockdown experiments. Transfection with 50 nM siRNA was repeated after the initial 24 h incubation. Cells were harvested 72 h following the initial transfection. Extracts for protein analyses were prepared by collecting cells in immunoprecipitation (IP) buffer (10 mM Tris (pH 7.4), 150 mM NaCl, 1 mM EGTA, 1 mM EDTA, 1% Triton X-100, 0.5% IGEPAL CA-630, protease inhibitors (1 mM phenylmethylsulfonyl fluoride, 1 µM pepstatin, 50 trypsin inhibitory milliunits of aprotinin, 10 µM leupeptin, 1 mM 1,10-phenanthroline), and phosphatase inhibitor (0.2 mM sodium vanadate)). To extract RNA for gene expression analyses, cells were harvested in buffer provided in the RNeasy mini kit (Qiagen, Hilden, Germany). As a positive control to assess siRNA transfection efficiency, a knockdown of Cyclophilin B (Dharmacon, Lafayette, CO, USA; Cat #: D-001820-02-05) was also performed [28]. 

### 2.3. RNA Analysis 

Total RNA was extracted, isolated, and reverse transcribed (RT), and cDNA was quantified using real-time quantitative PCR (qPCR) as previously described [28]. Using an Applied Biosystems 7900HT Fast qPCR System and the amplification program specified in the TB Green master mix kit (Clontech, Mountain View, CA, USA; Cat #: RR420A), expression levels of the following murine genes were quantified: *Ehmt2(G9a)*, *Ehmt1(GLP)*, Adiponectin, *Lcn2*, *Il-6*, *Mcp1*, *Glut4*, *Atgl*, *Hsl*, and *Plin1.* The reference genes were Cyclophilin B, Ubiquitin B, and *Nono.* Primers (sequences shown in Table 1 along with full gene names and) were purchased from Integrated DNA Technologies (IDT, Coralville, IA, USA). 

### 2.4. Whole-Cell Extract Preparation and Subcellular Fractionation

The preparation of whole-cell extracts and subcellular fractionation were performed as previously described [27,28]. Whole-cell extracts were prepared in either IP or RIPA buffer. Prior to subcellular fractionation, mature 3T3-L1 adipocytes (twelve 10-cm plates per treatment group) were treated with vehicle or murine TNFα for a specified amount of time as indicated in the figure legend. The protein content of whole-cell extracts and subcellular fractions was quantified using the Bicinchoninic acid (BCA) assay kit (Sigma–Aldrich, St. Louis, MO, USA; Cat #: BCA1).

### 2.5. Gel Electrophoresis and Immunoblotting

The samples were separated on 6%, 7.5%, 10%, or 15% sodium dodecyl sulfate (SDS) polyacrylamide (PA) gels and transferred to nitrocellulose membranes that were blocked in 5% milk as previously described [28]. Enhanced chemiluminescence signals produced by horseradish peroxidase-conjugated secondary antibodies were visualized on X-ray film [28].

### 2.6. Immunoprecipitation (IP) 

Cell extracts (300 µg total protein) diluted in IP buffer were incubated overnight with 5 µg of target antibody. Conjugation to Protein A-beads for 4 h, washes, and elution into SDS loading buffer (LB) were performed as previously described [28]. LB-extracted IP supernatants were analyzed using SDS-PAGE and immunoblotting. The negative control for each IP experiment (mock sample) contained only IP buffer (no cell extract) and the IP antibody. 

### 2.7. Measurement of Glycerol and Free Fatty Acid Release

Fifty-four hours following siRNA transfection as described in Section 2.2, the medium was changed to 1% calf/DMEM and the cells were treated with an equivalent volume of vehicle (0.1% BSA/PBS) or 0.75 nM TNFα overnight (16 h). As previously described [28], following incubation, the treatment medium was replaced with incubation medium containing fresh vehicle or TNFα. Glycerol and free fatty acid lipolysis products were allowed to accumulate for 2 h before the conditioned medium was collected and stored at −20 °C. Using 25 μL of conditioned medium from each well, glycerol or nonesterified free fatty acid (NEFA) was quantified according to the respective assay kit protocol (glycerol: Millipore, Burlington, MA, USA; Cat #: OB100; NEFA: Wako Diagnostics, Mountain View, CA, USA; HR Series NEFA-HR(2) Color Reagent A 999-34691, HR Series NEFA-HR(2) Solvent A 995-34791, HR Series NEFA-HR(2) Color Reagent B 991-34891, HR Series NEFA-HR(2) Solvent B 993-35191).

### 2.8. Antibodies

Anti-G9a (PP-A8620A-00; mouse monoclonal) and anti-GLP (PP-B0422-00; mouse monoclonal) antibodies were purchased from R&D Systems (Minneapolis, MN, USA). Anti-H3K9me1 (ab8896; rabbit polyclonal) and anti-STAT5A (ab32043; rabbit monoclonal) antibodies were purchased from Abcam (Cambridge, UK). Anti-Adiponectin (PA1-054; rabbit polyclonal) antibody was purchased from Thermo Scientific (Waltham, MA, USA). Anti-H3K9me2 (9753; rabbit polyclonal), anti-Histone H3 (14269; mouse monoclonal), anti-HSL (4107; rabbit polyclonal), anti-Perilipin (3470; rabbit polyclonal), and anti-β-actin (3700; mouse monoclonal) antibodies were purchased from Cell Signaling (Danvers, MA, USA).

### 2.9. Statistical Analysis 

Data from cultured adipocyte experiments are shown as mean ± standard error of the mean (SEM) in bar graphs and were analyzed using a two-tailed unpaired Student’s *t*-test (GraphPad Prism 6.01). A cutoff of *p* < 0.05 was used to consider results statistically significant.

## 3. Results

In mice, G9a and GLP expression in adipocytes contributes to metabolic health, and loss of expression of either one of these methyltransferases is associated with systemic metabolic dysfunction [13,16]. Hence, we hypothesized that the loss of G9a or GLP would have cell-autonomous effects in cultured mature adipocytes. We performed siRNA experiments to knockdown either G9a or GLP in murine 3T3-L1 adipocytes to determine if the expression of either protein impacted H3K9 methylation and/or TNFα-regulated gene expression. As shown in Figure 1, we effectively knocked down the protein expression of G9a without significantly modulating GLP levels. In addition, we knocked down GLP protein expression without significantly affecting the protein expression of G9a (Figure 2).

G9a and GLP have been shown to be the primary methyltransferases for the mono- and di-methylation of H3K9 [1,29]. The mono-methylation of H3K9 is usually associated with transcriptional activation and the di-methylation of H3K9 is typically associated with transcriptional repression [30]. Examination of these two histone marks revealed that the siRNA-mediated knockdown of G9a did not have a significant effect on H3K9 mono-methylation in mature adipocytes under basal (vehicle) conditions (Figure 1). Yet, there was a trend towards decreased di-methylation of H3K9 following TNFα treatment in adipocytes lacking G9a (Figure 1). Our observations were modestly different in adipocytes lacking GLP. The loss of GLP did not affect the di-methylation of H3K9, but there was decreased mono-methylation in the presence of TNFα (Figure 2). As expected, TNFα exposure reduced adiponectin levels, and this was largely unaffected or exacerbated by the loss of either G9a or GLP (Figure 1 and Figure 2). The levels of total Histone 3 were assessed and used as a loading control since its expression levels were unchanged by any of the treatment conditions. TNFα increases lipolysis in adipocytes, but loss of either G9a or GLP expression did not affect TNFα-induced lipolysis as judged by glycerol release (Figure 3). Additionally, the loss of either G9a or GLP did not affect the ability of TNFα to regulate the gene expression of *Lcn2*, *Il6*, *Mcp1*, Adiponectin, and *Glut4*.

Previous studies have shown that G9a and GLP have to be in a complex in order to exert their catalytic activity [29], but data also indicate that the formation and activity of this complex is tissue-specific and dependent upon the developmental stage [2,13,16,31]. To our knowledge, it is not known if G9a and GLP physically interact in adipocytes. We performed coimmunoprecipitation with a G9a antibody followed by Western blotting using an anti-GLP antibody in the absence or presence of an acute thirty-minute TNFα treatment of 3T3-L1 adipocytes. Freshly isolated whole-cell extracts (WCE) and WCE that had undergone one freeze–thaw cycle were used in our studies. As shown in Figure 4, G9a interacts with GLP under basal (vehicle) and TNFα-treated conditions in adipocytes. The signal transducer and activator of transcription 5A (STAT5A) did not interact with G9a and was assessed as a negative control to show the specificity of the interaction between G9a and GLP (Figure 4). 

Since we observed that the knockdown of either G9a or GLP did not have prominent effects on adipocytes, we used siRNA to simultaneously knockdown both G9a and GLP protein expression in 3T3-L1 adipocytes. As shown in Figure 5, we knocked down *G9a* and *Glp* mRNA and protein expression. As observed with single knockdowns (Figure 1 and Figure 2), the siRNA-mediated knockdown of both G9a and GLP protein levels did not have a significant effect on the mono- and di-methylation of H3K9 under basal conditions. However, adipocytes lacking both G9a and GLP and treated with TNFα had a decrease in the mono- and di-methylation of H3K9 expression (Figure 5B,C). In basal conditions, there was an increase in adiponectin protein expression in G9a/GLP-knockdown adipocytes, but the TNFα suppression of adiponectin expression was not changed in adipocytes lacking these two methyltransferases. The levels of histone H3 were examined to demonstrate the even loading of protein samples and that the observed changes in histone methylation were not due to alterations in protein levels (Figure 5B). Interestingly, qPCR analysis revealed that knocking down both G9a and GLP in fully differentiated adipocytes increased *Adiponectin* and *Glut4* mRNA expression under basal conditions (top portion of Figure 6). In double-knockdown adipocytes, TNFα resulted in the expected decrease in *Adiponectin* and *Glut4* mRNA expression, and these effects were largely unaffected by the double knockdown. Notably, the loss of G9a and GLP expression did not affect the basal levels of *Lcn2*, *Il6*, and *Mcp1* mRNA expression in adipocytes but did enhance the ability of TNFα to stimulate the expression of these proinflammatory genes (bottom portion of Figure 6). 

Since we observed that the loss of both G9a and GLP levels affected the ability of TNFα to induce proinflammatory gene expression (Figure 6), we also assessed whether G9a and GLP expression affected TNFα-induced lipolysis. Lipolysis is a highly regulated metabolic process that involves the breakdown of triglycerides into fatty acids and glycerol, which are subsequently released from adipocytes. As shown in Figure 7, TNFα-induced lipolysis was assessed via measurements of both glycerol and free fatty acid release into the culture medium. A double knockdown of G9a and GLP significantly enhanced the effect of TNFα on lipolysis but did not impact basal lipolysis (Figure 7). 

Although TNFα increases lipolysis, it has also been shown to reduce expression of mRNA or proteins that promote lipolysis, such as adipose triglyceride lipase (ATGL), hormone sensitive lipase (HSL), and perilipin 1 (PLIN1) [32,33]. We examined the mRNA and protein expression of these lipolytic mediators and observed that the loss of both G9a and GLP increases *Atgl*, *Hsl*, and *Plin1* mRNA expression under basal and TNFα-stimulated conditions, but this does not result in increased protein levels (Figure 8). TNFα treatment substantially reduced the expression of ATGL, HSL, and PLIN1 protein levels, but there were no significant changes between G9a/GLP-knockdown adipocytes and control adipocytes (Figure 8).

RelA (also known as p65) is the major functional subunit of the transcription factor nuclear factor kappa-light-chain-enhancer of activated B cells (NF-κB). p65 mediates some actions of TNFα and has a prominent role in inflammation. Since *Lcn2*, *Il6*, and *Mcp1* are p65 target genes induced by TNFα [20,21,34], we hypothesized that G9a and GLP would be present in a protein complex with p65. To test this hypothesis, we performed coimmunoprecipitation (co-IP) using an anti-G9a antibody and immunoblotted with an anti-p65 antibody. We analyzed both cytosolic and nuclear extracts prepared from mature 3T3-L1 adipocytes following an acute thirty-minute vehicle (V) or TNFα treatment. TNFα induced p65 translocation to the nucleus as shown in the direct western portion of Figure 9A. A mock sample containing anti-G9a antibody without cell extract was used as a negative control. The G9a antibody efficiently pulled down G9a in the nucleus under both vehicle and TNFα conditions but was able to coimmunoprecipitate p65 only under TNFα-stimulated conditions (Figure 9A). Notably, these experiments were performed in nontransfected adipocytes with endogenous proteins. The G9a antibody was not able to co-IP STAT5A under any of the conditions tested, and this result served as a negative control to show the specificity of the interaction between G9a and p65. The majority of STAT5A is present in the cytosol and its location is not regulated by TNFα. These observations are consistent with published data that TNFα does not activate and induce the translocation of STAT5A to the nucleus in 3T3-L1 adipocytes [35]. 

To further validate that G9a and GLP proteins form a complex with p65 in adipocytes, we demonstrated that nuclear p65 could also be coimmunoprecipitated by an anti-GLP antibody (Figure 9B). Together, these novel observations demonstrate that G9a and GLP are present in a protein complex with p65 under TNFα-stimulated conditions in the nucleus of 3T3-L1 adipocytes. 

## 4. Discussion

It is well-established that G9a and GLP are the primary methyltransferases for the mono-and di-methylation of H3K9 in many cell types [31,36,37,38,39]. Previous studies observed that the deletion of G9a resulted in a robust reduction in the di-methylation of H3K9 suggesting that G9a is the primary methyltransferase in the di-methylation of H3K9 in primary murine brown preadipocytes [13]. To our knowledge, our study is the first to examine the effects of G9a and GLP on the mono- and di-methylation of H3K9 in fully differentiated 3T3-L1 “white” adipocytes under basal and TNFα-stimulated conditions. We were surprised to observe that the loss of either G9a or GLP did not substantially regulate these two histone H3 epigenetic marks in fully differentiated adipocytes (Figure 1 and Figure 2). These studies indicate that G9a or GLP alone are not the sole regulators of H3K9 methylation in mature adipocytes, but there is also evidence that the mono- and di-methylation of H3K9 can be modulated by other methyltransferases including SETDB1 and SUV39H1 [40,41,42]. Hence, it is plausible that knocking down either G9a or GLP results in compensation by another H3K9 methyltransferase. There is evidence that G9a, GLP, SUV39H1, and SETDB1 can form a multimeric complex and cooperate to affect H3K9 methylation in euchromatin and heterochromatin [43]. Collectively, these results indicate that the methylation of H3K9 depends on cell type and/or differentiation status and underscores the complexity of epigenetic regulation. 

Since H3K9 methylation was not substantially altered by our single knockdowns of G9a and GLP methyltransferases, we performed a double knockdown of G9a and GLP. We consistently observed that the simultaneous loss of G9a and GLP resulted in increased Adiponectin and *Glut4* gene expression under basal (vehicle treatment) conditions (Figure 6). TNFα can repress Adiponectin gene expression in mouse and human adipocytes [44,45], and in our studies we observed TNFα-induced decreases in adiponectin gene and protein expression under both the control and G9a/GLP KD conditions. However, it was surprising that the mono- and di-methylation of H3K9 was only significantly affected by the loss of both G9a and GLP when adipocytes were treated with TNFα, but not under basal conditions (Figure 5). This may suggest that the observed increase in Adiponectin and *Glut4* gene expression is not dependent on G9a and GLP’s ability to methylate H3K9 and most likely results from the loss of methylation of a nonhistone substrate. There is significant data in other model systems to demonstrate that G9a and GLP have nonhistone substrates and can act independently of their enzymatic activity [5,6,46]. 

Increased GLUT4 expression [23,47] and increased adiponectin levels [48] in adipocytes are associated with improvements in insulin sensitivity. Previous studies have suggested a role of G9a in hepatic insulin signaling [3], but it is not known if G9a and/or GLP impact insulin signaling in white adipocytes. Our data suggest that G9a and GLP may regulate adipocyte insulin sensitivity under basal conditions due to their ability to alter Adiponectin and *Glut4* expression. However, additional experiments will be needed to determine if G9a/GLP expression in adipocytes is metabolically favorable by directly assessing if the knockdown and/or overexpression of G9a and GLP expression affects insulin signaling in adipocytes. 

We observed that the TNFα stimulation of *Lcn2*, *Il6*, and *Mcp1* gene expression was significantly increased in G9a/GLP-deficient adipocytes compared to control adipocytes (Figure 6), which suggests that G9a and GLP expression may have a role in suppressing the ability of TNFα to induce genes associated with inflammation. Enriched H3K9 di-methylation in the transcriptional start sites of genes correlates with gene silencing. Thus, the observed reduction in this histone mark could contribute to increased expression of proinflammatory genes (*Lcn2*, *Il6*, and *Mcp1*) in TNFα-stimulated adipocytes with depleted G9a/GLP. Future studies will be necessary to determine if the loss of G9a and GLP expression reduces the di-methylation of H3K9 at *Lcn2*, *Il6* and/or *Mcp1*-specific gene promoter regions in response to TNFα. In Figure 4, we show that G9a and GLP interact under both basal and TNFα-stimulated conditions, and this indicates that molecular mechanisms other than increased G9a/GLP complex formation play a role in regulating TNFα responses in fully differentiated white adipocytes. 

NF-κB is a primary transcription factor that mediates the action of TNFα as well as other proinflammatory cytokines. TNFα stimulates the translocation of the p65 subunit of NF-κB to the nucleus where it directly binds to NF-κB sites in the promoters of proinflammatory genes such as *Lcn2*, *Il6*, and *Mcp1* [20,34]. Since G9a and GLP play a role in inflammation in immune cells and cancer cells via interactions with members of the NF-κB family (p65, p50, and RelB) [49,50,51], we investigated the possibility of a similar molecular mechanism occurring in adipocytes. We observed that p65 only associates with G9a and GLP under TNFα-stimulated conditions (Figure 9). Although controversial, increased circulating LCN2 is typically associated with insulin resistance and increased inflammation [52,53]. There is also emerging evidence that the p65 subunit of NF-κB can be regulated through methylation on various lysine residues [54]. Studies suggest that lysine methyltransferases are recruited to specific promoter regions after NF-κB translocates to the nucleus and binds to its target genes [54]. Once methyltransferases are recruited to DNA, they can methylate lysine residues on nearby histones as well as NF-κB [54]. Since we observed that p65 only associates with G9a and GLP under inflammatory conditions, we hypothesize that the G9a/GLP complex regulates the transcriptional activity of p65 through the methylation of p65 itself in response to TNFα in adipocytes.

TNFα is a potent inducer of lipolysis and has been shown to increase basal lipolysis in adipocytes isolated from human subcutaneous adipose tissue (SAT) [33]. However, the underlying mechanism of TNFα-induced lipolysis remains unknown. It typically takes greater than six hours to detect TNFα-induced changes in lipolysis in 3T3-L1 adipocytes, which suggests that this regulation is largely mediated by genomic effects. Experiments performed in human adipocytes demonstrated that the NF-κB signaling pathway is important for the lipolytic response to TNFα [55]. As shown in Figure 7, the loss of G9a and GLP increases TNFα-induced lipolysis suggesting that the loss of G9a and GLP in adipocytes may contribute to the metabolically unhealthy phenotype observed in G9a and GLP adipocyte-specific knockout mice [13,16]. Enhanced lipolysis increases the circulating levels of glycerol/free fatty acids—a clinical manifestation that is considered harmful and is typically associated with metabolically unhealthy obesity and insulin resistance [56]. Additionally, elevated free fatty acids in circulation can lead to ectopic storage in skeletal muscle and the liver [57]. Our results show an increase in glycerol and free fatty acid release from TNFα-stimulated G9a/GLP-deficient adipocytes that occurs in a cell-autonomous manner and suggest that G9a and GLP play a role in restricting TNFα’s ability to induce lipolysis. Since TNFα affects the expression of proteins known to regulate lipolysis [58], we examined if G9a/GLP knockdown had an effect on the gene or protein expression of ATGL, HSL, and PLIN. As previously documented, TNFα robustly reduced the mRNA expression of *Atgl*, *Hsl*, and *Plin1* [32,33]. Interestingly, adipocytes with reduced levels of G9a and GLP had increased *Atgl*, *Hsl*, and *Plin1* mRNA expression independent of TNFα treatment (Figure 8). However, this effect did not translate into similar changes in protein expression suggesting that TNFα-induced lipolysis is not due to the increased expression of these three fundamental prolipolytic proteins. Further studies are needed to determine the specific mechanism of G9a and GLP’s ability to regulate TNFα-induced lipolysis.

The majority of the data presented were collected by Dr. Able as a part of her doctoral research project, and as such the results and discussion have been previously published in her doctoral dissertation [26].

## 5. Conclusions

Under basal conditions, the loss of G9a and GLP results in a metabolically beneficial phenotype as judged by increased Adiponectin and *Glut4* gene expression (Figure 6). Yet, in the presence of a cytokine that promotes metabolic dysfunction, the loss of these methyltransferases increases TNFα action to promote both proinflammatory gene expression and lipolysis (Figure 6 and Figure 7). Clearly, these novel observations warrant further research. Additional studies are needed to identify if other lysine methyltransferases are present in the G9a/GLP complex, to assess if G9a/GLP expression affects the expression of other TNFα-regulated genes in adipocytes, and to examine if our observations are reproducible in other cellular models such as primary murine adipocytes and human adipocytes. 

Overall, our novel data showing that the loss of both G9a and GLP exacerbates TNFα’s deleterious effects on inflammatory gene expression and lipolysis are consistent with the metabolically unfavorable phenotype observed in adipocyte-specific G9a and GLP knockout mice [13,16]. Our data indicate that G9a and GLP have a protective role in modulating the transcriptional activity of p65 in response to TNFα exposure. Although G9a and GLP expression levels impact TNFα-mediated gene expression changes and lipolysis, further studies are needed to determine whether these effects require the G9a/GLP-mediated methylation of H3K9 and/or nonhistone substrates. Additionally, future studies can determine whether these observations are specific to TNFα or whether they also apply to other proinflammatory cytokines. Overall, these studies underscore the complexity of epigenetic contributions to gene expression. 

## Figures and Tables

**Figure 1 biology-12-00674-f001:**
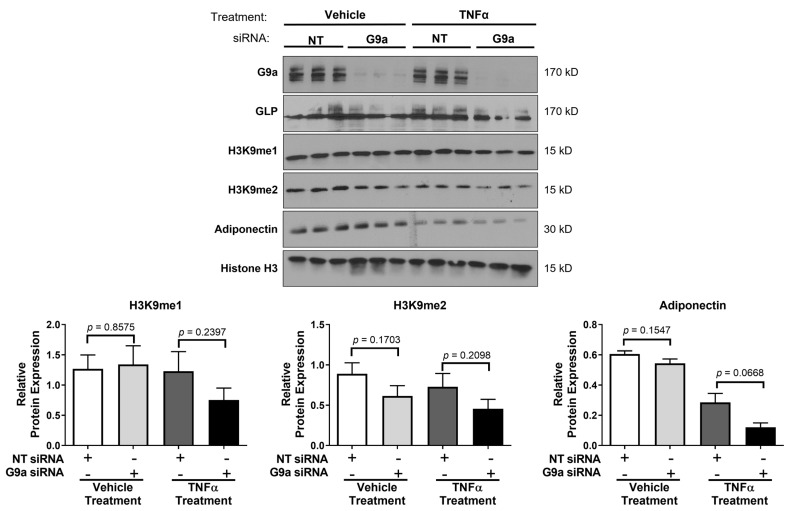
Knockdown of G9a does not have a significant effect on mono- and di-methylation of H3K9 in 3T3-L1 adipocytes. Fully differentiated 3T3-L1 adipocytes were transfected with nontargeting (NT) siRNA and G9a siRNA for 54 h. Then, cells in 10% FBS were treated with 0.75 nM TNFα or an equivalent volume of 0.1% BSA/PBS (vehicle). After 16 h, cells were again treated with TNFα or vehicle for 2 h. Monolayers were harvested for protein, and 25 μg of total protein from whole-cell extract samples was analyzed via immunoblotting using antibodies specified on the left in top panel. Three replicates for each treatment group as shown were used in each individual experiment. This experiment was independently performed three times on different groups of adipocytes, and a representative figure is shown. Western blot results were quantified using Image Studio Lite and normalized to Histone H3 expression as a loading control. The data are expressed as the mean ± SEM from three independent experiments. (See Appendix A for the original Western blot images).

**Figure 2 biology-12-00674-f002:**
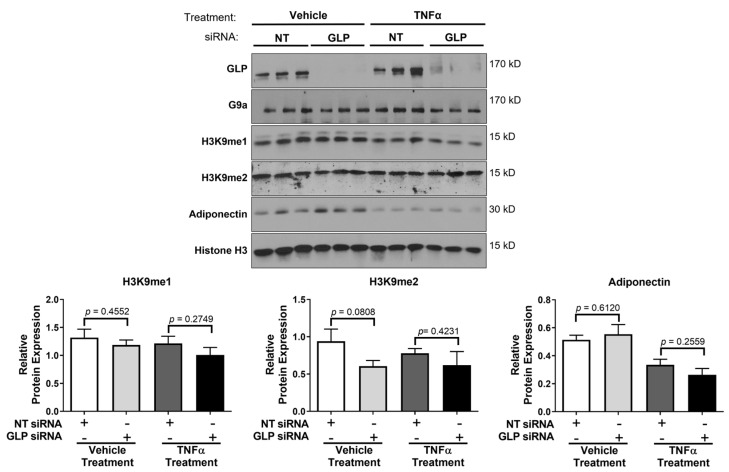
Knockdown of GLP does not have a significant effect on mono-and di-methylation of H3K9 in 3T3-L1 adipocytes. Knockdown of GLP, using GLP siRNA, and cell treatments were performed as described in the Figure 1 legend. Monolayers were harvested for protein, and 25 μg of total protein from whole-cell extract samples was analyzed via immunoblotting using antibodies specified on the left in top panel. Three replicates for each treatment group as shown were used in each individual experiment. This experiment was independently performed three times on different groups of adipocytes, and a representative figure is shown. Western blot results were quantified using Image Studio Lite and normalized to Histone H3 expression as a loading control. The data are expressed as the mean ± SEM from three independent experiments. (See Appendix A for the original Western blot images).

**Figure 3 biology-12-00674-f003:**
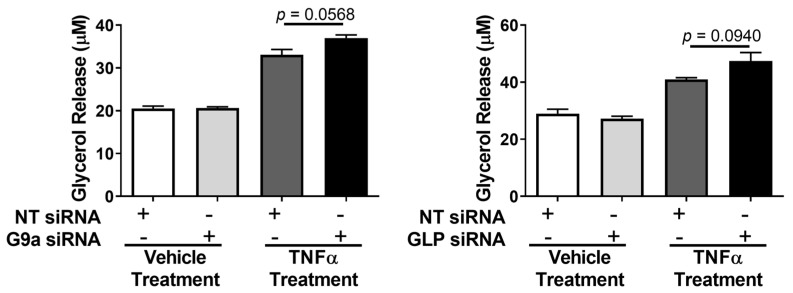
Knockdown of either G9a or GLP does not have a significant effect on basal or TNFα-induced glycerol release in 3T3-L1 adipocytes. Knockdown of G9a or GLP, using G9a or GLP siRNA, and cell treatments were performed as described in the Figure 1 legend. Glycerol content was measured using 25 μL of conditioned medium. Statistical significance was determined using an unpaired Student’s *t*-test. This is a representative figure of an experiment independently performed three times on different groups of adipocytes.

**Figure 4 biology-12-00674-f004:**
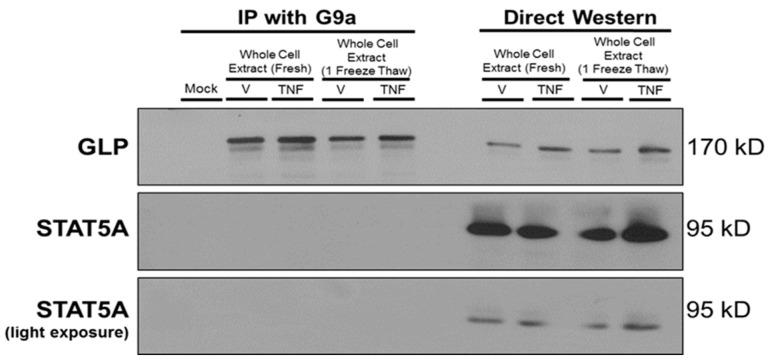
Endogenous G9a and GLP physically interact under basal and TNFα-stimulated conditions in 3T3-L1 adipocytes. Fully differentiated 3T3-L1 adipocytes were treated with 0.75 nM TNFα or an equivalent volume of vehicle (0.1% BSA/PBS) for 30 min. An amount of 200 μg total protein in whole-cell extract samples was immunoprecipitated (IP) with an anti-G9a antibody and analyzed using immunoblotting (left half of strips). The mock sample contained an anti-G9a antibody without cell extract and was used as a negative control. The right of each strip shows direct Western blot controls, in which 25 μg total protein was directly subjected to immunoblotting without IP. This is a representative figure of an experiment independently performed three times. (See Appendix A for the original Western blot images).

**Figure 5 biology-12-00674-f005:**
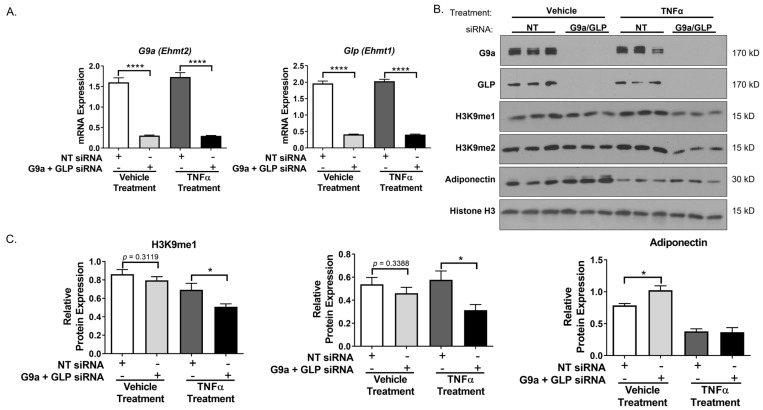
Simultaneous knockdown of G9a and GLP decreases mono- and di-methylation of H3K9 expression under TNFα-stimulated conditions in 3T3-L1 adipocytes. Knockdown of G9a and GLP, using both G9a and GLP siRNA, and cell treatments were performed as described in the Figure 1 legend. Monolayers were harvested for RNA or protein. (**A**) Total mRNA expression was measured using quantitative RT-PCR. Target gene expression was normalized to *Cyclophilin A*. (**B**) 25 μg of total protein from whole-cell extract samples was analyzed via immunoblotting using the antibodies specified on the left panel. Histone H3 was used as a loading control. (**C**) Western blot results were quantified using Image Studio Lite and normalized to Histone H3 expression. Three replicates were used for each treatment group in each individual experiment. These are representative figures of experiments independently performed three times on different groups of adipocytes. Quantitative data are expressed as the mean ± SEM from three independent experiments. Statistical significance was determined using an unpaired Student’s t-test and assigned as * *p* < 0.05 or **** *p* < 0.0001. (See Appendix A for the original Western blot images).

**Figure 6 biology-12-00674-f006:**
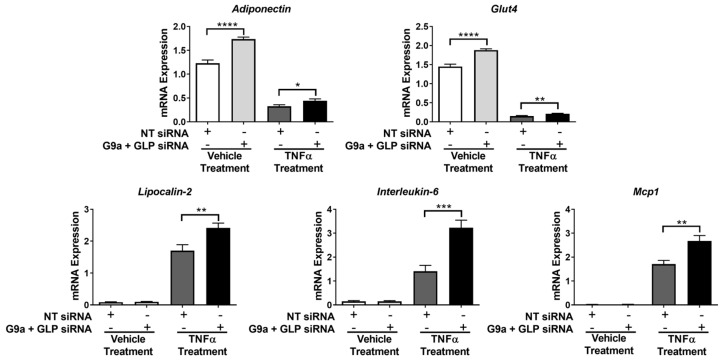
Simultaneous knockdown of G9a and GLP increases Adiponectin and *Glut4* gene expression under basal conditions and significantly enhances TNFα’s ability to induce proinflammatory gene expression in 3T3-L1 adipocytes. Knockdown of G9a and GLP, using both G9a and GLP siRNA, and cell treatments were performed as described in the Figure 1 legend. Monolayers were harvested for RNA, which was subjected to quantitative RT-PCR to determine gene expression levels. Expression of each gene, represented by an individual graph, was normalized to Cyclophilin A. Three replicates were used for each treatment group in each individual experiment. This is a representative figure of an experiment independently performed three times on different groups of adipocytes. The data are expressed as the mean ± SEM from three independent experiments. Statistical significance was determined using an unpaired Student’s t-test and assigned as * *p* < 0.05, ** *p* < 0.01, *** *p* < 0.001, and **** *p* < 0.0001.

**Figure 7 biology-12-00674-f007:**
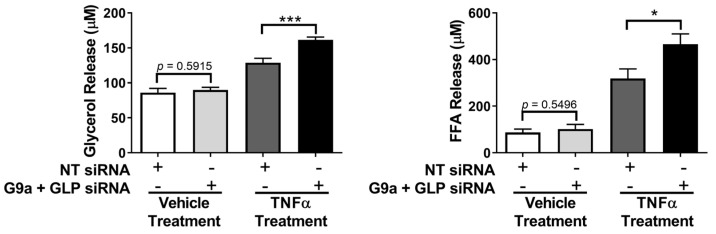
Simultaneous knockdown of G9a and GLP increases the ability of TNFα to induce lipolysis in 3T3-L1 adipocytes. Knockdown of G9a and GLP, using both G9a and GLP siRNA, and cell treatments were performed as described in the Figure 1 legend. Glycerol and free fatty acid (FFA) levels were measured using 25 μL of conditioned medium per assay. This is a representative figure of an experiment independently performed two times on different groups of adipocytes. The data are expressed as the mean ± SEM from three independent experiments. Statistical significance was determined using a Student’s *t*-test and assigned as * *p* < 0.05 and *** *p* < 0.001.

**Figure 8 biology-12-00674-f008:**
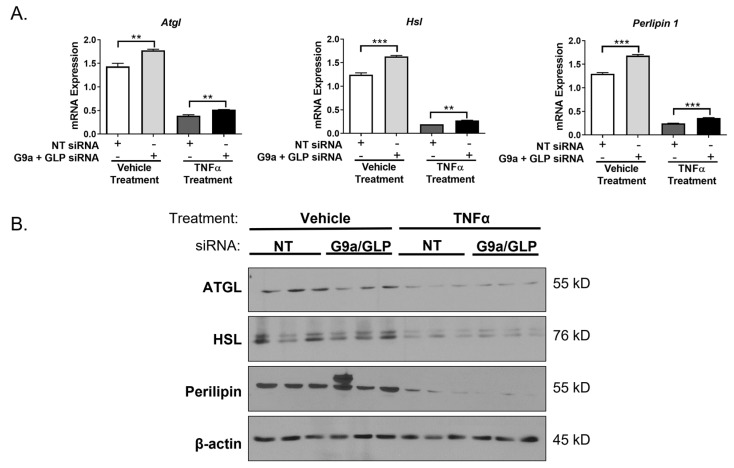
Simultaneous knockdown of G9a and GLP increases lipolytic gene expression but does not affect lipolytic protein expression. Knockdown of G9a and GLP, using both G9a and GLP siRNA, and cell treatments were performed as described in the Figure 1 legend. Monolayers were harvested for RNA and protein. (**A**) Total mRNA expression was measured using quantitative RT-PCR. Expression of each gene, represented by an individual graph, was normalized to *Cyclophilin A*. Statistical significance was determined using a Student’s t-test and assigned as ** *p* < 0.01 or *** *p* < 0.001. (**B**) 25 μg of total protein was analyzed via immunoblotting using the antibodies specified on the left of each strip. Three biological replicates per treatment group were included in each individual experiment. This is a representative figure of an experiment independently performed three times on different groups of adipocytes. (See Appendix A for the original Western blot images).

**Figure 9 biology-12-00674-f009:**
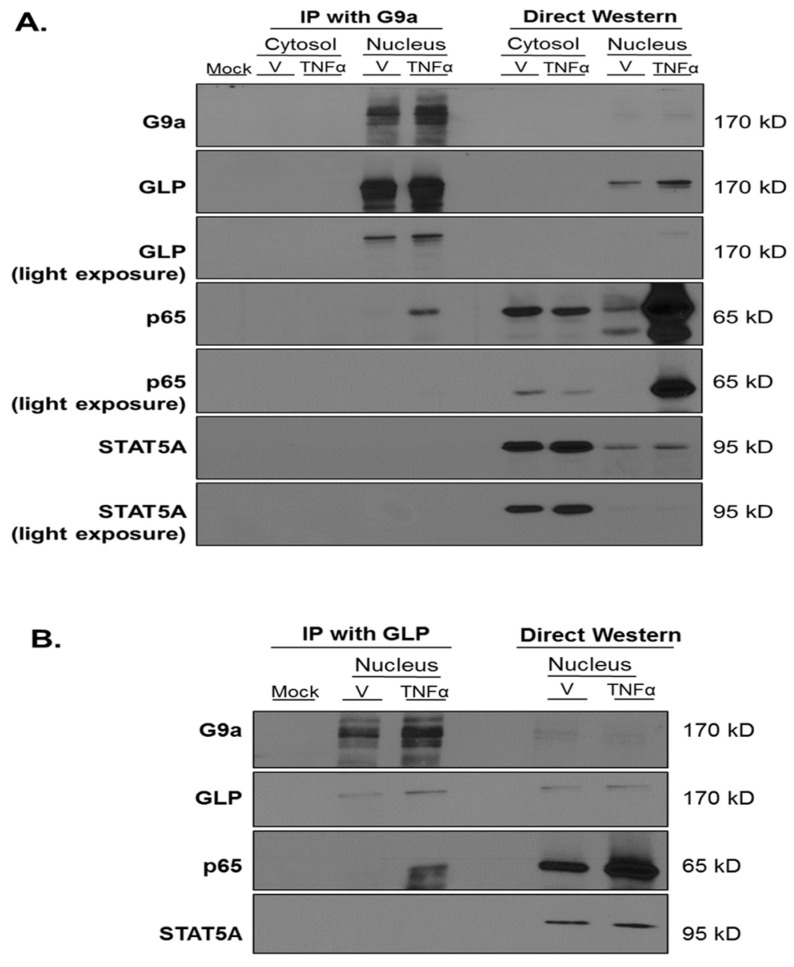
p65 is present in a complex with G9a and GLP under TNFα-stimulated conditions in 3T3-L1 adipocytes. Fully differentiated 3T3-L1 adipocytes were treated with 0.75 nM TNFα or an equivalent volume of vehicle (0.1% BSA/PBS) for 30 min. (**A**) Monolayers were collected and subjected to subcellular fractionation. As shown in the left-hand portion of the figure, cytosolic and nuclear protein extracts (300 μg total protein/sample) were immunoprecipitated (IP) with an anti-G9a antibody. The mock sample contained anti-G9a antibody without cell extract and was used as a negative control. (**B**) The left-hand portion of the figure shows IP of nuclear extract (300 μg total protein/sample) using anti-GLP antibody. The mock sample contained anti-G9a antibody (**A**) or anti-GLP antibody (**B**) without protein extract and was used as a negative control in each experiment. The samples on the right-hand portion of (**A**) or (**B**) are direct western blot controls containing 25 μg total protein that were analyzed via immunoblotting without IP. This is a representative figure of an experiment independently performed three times. (See Appendix A for the original Western blot images).

**Table 1 biology-12-00674-t001:** Primer sequences of each gene used in qPCR.

Gene; Abbreviation (GenBank Accession No.)	Primer 1 Sequence	Primer 2 Sequence
Cyclophilin A; *PpiA* (NM_008907)	5′-CCACTGTCGCTTTTCGCCGC-3	5′-TGCAAACAGCTCGAAGGAGACGC-3′
Cyclophilin B; *PpiB* (NM_0011149)	5′-CCGTAGTGCTTCAGCTTGA-3′	5′-AGCAAGTTCCATCGTGTCATC-3′
Ubiquitin B; *Ubb* (NM011664)	5′-GCTTACCATGCAACAAAACCT-3′	5′-CCAGTGGGCAGTGATGG-3′
Non-POU domain containing octamer binding; *Nono* (NM_001252518)	5′-TCTTCAGGTCAATAGTCAAGCC-3′	5′-CATCATCAGCA TCACCACCA-3′
*G9a/Ehmt2* (NM_001286573)	5′-TCCTCCTCACTCAACTGTTCA-3′	5′-CGATGACTTCAGCCTGT ACTATG-3′
*Glp/Ehmt1* (NM_001012518)	5′-TCCATCAACCAGCATGAGAAG-3′	5′-GTGCTCTAATCGCTCTAGACTC-3′
suadAdiponectin (NM_009605)	5′-GCAGGA TTAAGAGGAACAGGAG-3′	5′-TGTCTGTACGATTGTCAGTGG-3′
Lipocalin 2; *Lcn2* (NM_008491)	5′-AGTCACATTCGTTGCAGAAGA-3′	5′-CAGAGATGTGCCTCCA TACTG-3′
Interleukin 6; *Il6* (NM_031168)	5′-AGTACATCTCCAGTCTCCTCAG-3′	5′-ATGCTCTTCAGTTCGTGTGT-3′
Monocyte chemoattractant protein 1; *Mcp1/Ccl2* (NM_011333)	5′-GCAGAGAGCCAGACGGGAGGA-3′	5′-TGGGGCGTTAACTGCATCTGG-3′
Glucose transporter 4; *Glut4/Slc2a4* (NM_009204)	5′-TCTT A TTGCAGCGCCTGAG-3′	5′-GAGAATACAGCT AGGACCAGTG-3′
Adipose triglyceride lipase; *Atgl/Pnpla2* (NM_001163689)	5′-CTCATAAAGTGGCAAGTTGTCTG-3′	5′-GAGCTCATCCAGGCCAA T-3′
Hormone sensitive lipase; *Hsl/Lipe* (NM_010719)	5′-CTCGTTGCGTTIGTAGTGC-3′	5′-CTGCAAGAGTATGTCACGCTA-3′
Perilipin 1; *Plin1* (NM_175640)	5′-CGTGGAGAGTAAGGATGTCAATG-3′	5′-GTGCTGTTGTAGGTCTTCTGG-3′

## Data Availability

Some or all data generated or analyzed during this study are included in this published article.

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
