# Peer review of "TNFα Effects on Adipocytes Are Influenced by the Presence of Lysine Methyltransferases, G9a (EHMT2) and GLP (EHMT1)"

_biology, 2023, doi:10.3390/biology12050674_

Round 1
Reviewer 1 Report
The goal of this study is to examine the role of the lysine methyltransferases G9a and GLP in adipocytes and examine the consequences of modulating their action on fat cell function in response to proinflammatory signals from TNFa. Overall this is an interesting study, and the experiments are carefully performed and clearly presented.
Specific comments:
1. The manuscript might be easier to understand with some reorganization. Figures 1 and 2 show that single loss of G9a or GLP do not alter histone modification or adiponectin levels in response to TNFa. Similarly Figure 3 shows that single loss of these factors does not alter TNF-mediated glycerol release. Since the design of the experiments are very similar (Figures 1, 2, 3, 5), it would improve the clarify of study to assemble the studies with single knockdown with the simultaneous KD (Fig 5B). Alternatively the single knockdown studies could be presented as supplemental data and then focus the work on the simultaneous knockdown.
2. Since single loss of function in adipocytes promotes a metabolically unfavorable phenotype, it is not clear how these findings in cells models relate. If the dimer of G9a and GLP is the functional entity, then it is surprising why individual knockdown did not have a phenotype. Related to this question, the dynamics of H3K9me1 and HSK9me2 could be explored with additional time points, is it just that these modifications are stable in the conditions studied. Since others have shown that single loss results in cellular changes, is this observation cell type dependent and that the 3T3-L1s used in this study have different complexes or compensation mechanisms?
3. The concept that G9a and GLP have non-histone substates is interesting. Additional information on establishing the importance of non-histone modifications is needed. While it is mentioned in the discussion that “Adiponectin and Glut4 gene expression is not dependent on G9a and GLP’s H3K9 methyltransferase activity and most likely results from loss of methylation of a non-histone substrate”. Additional experimental data to support this claim would significantly strengthen the study and add to its novelty and interest. For example, can mutations in the potential methylation sites in non-histone targets be generated to test their function?
4. Figure 4. G9a is missing to show efficiency of IP
5. Figure 8. Efficiency of KD is needed. Explanation of perilipin doublet.
Reviewer 2 Report
Manuscript review response: “TNFα effects on adipocytes are influenced by the presence of 2 lysine methyltransferases, G9a (EHMT2) and GLP (EHMT1)”.
Manuscript ID: biology-2289197
Comments
It is an interesting study about epigenetic mechanisms in adipocytes, that involve the protective role of EHMT2 and EHMT1 before environment inflammatory generated by TNF.
Line 8 add LPG abbreviation.
Line 11 you mentioned that “G9a and 10 GLP are important in the development of metabolic disease” are important proteins, enzymes or what?
In your introduction you can write about the importance of epigenetic in obesity.
Line 39 you could be start with the characteristic and uses of 3T3-L1 cells. Why is important explore cellular mechanisms in these kinds of cells? is a model of diabetes or obesity and metabolic syndrome.
Line 58 you mentioned “However, experiments using a G9a/GLP specific inhibitor that suppresses its catalytic function, BIX-01294, had no effect on IRα and phosphorylated AKT levels”, is important add in what kind of animal model, clinical study or what kind of cellular line found these results.
Line 77-79 this paragraph “Considering G9a and GLP expression in adipocytes contributes to metabolic health in mice, we designed our experiments to explore if loss of G9a and GLP could modulate the effect of tumour necrosis factor alpha (TNFα) action in adipocytes”, I suggest to change before of materials and methods, because is your principal aim.
Line 91-104 I suggest al this paragraph is necessary rewriter because the writer it's a little confusing in the current version.
Line 107-114 You need to mention the plates you used and the number of cells you placed per well.
Line 142 could you add in the table 1 the gene accession or ID of gene bank basis in the primers that you used.
Line 156-158 you need add a best description of the western blot, you did not mention which was the staining method and what equipment used.
Line 222-233 you could place parentheses in each graph and add in the writing of figure 1, 2, 3,4, 5 and 6.
Line 338-348 you could place parentheses in each graph and add in the writing of figure 7.
Line 364-367 You could homogenize A. in the figure or A) in the writing. You could add a subsection or parenthesis to each graphic of the gene that was expressed and explain it in the writing of figure 8.
Line 416-417 what is your opinion about these findings? I think is necessary write a supposed about this or add an image representing this hypothesis in 3T3-L1 cells.
Line 437-439 you have explored the role of PPAR alpha and its participation in adiponectin secretion in this kind of cells?
Line 448-459 it is important to define the possible participation role of G9a and GLP, if it is protective or not, in the write of manuscript.
Line 475 the paragraph “under TNFα stimulated conditions” I think is better mentioned “under inflammatory stimuli conditions”.
Conclusion
Line 513-516 I suggest all this information “Overall, our novel data showing that loss of both G9a and GLP enhances TNFα’s ability to induce pro-inflammatory gene expression and lipolysis are consistent with the metabolically unfavourable phenotype observed in G9a and GLP adipocyte-specific KO 515 mice”, you could be change in discussion section.
Line 516-524 in this section of the conclusions you could be more concise in your writing of findings.
Line 516-517 The principal role of G9a and GLP in adipocytes is protective?
Author Response
We appreciate the feedback from the three reviewers on our submitted manuscript “TNFα effects on adipocytes are influenced by the presence of lysine methyltransferases, G9a (EHMT2) and GLP (EHMT1)”. We are thankful that it was indicated that “this is an interesting study, and the experiments are carefully performed and clearly presented. We have addressed all the reviewers’ comments, but do not have the ability to perform additional experiments. Also, we are trying to meet the April 6 deadline from the Editor. Thank you for your consideration of our revised manuscript to Biology (Manuscript ID: biology-2289197).
In addition to the point-by-point responses below, we have attached a marked version of our revised manuscript.
Reviewer 2 Comment: Line 8 add LPG abbreviation.
Response: We have now defined GLP as G9a-like protein.
Reviewer 2 Comment: Line 11 you mentioned that “G9a and 10 GLP are important in the development of metabolic disease” are important proteins, enzymes or what?
Response: We revised the statement as follows: “Studies show that G9a and GLP are important proteins in the development of metabolic disease, but their role in fat cell function is not well understood.”
Reviewer 2 Comment: In your introduction, you can write about the importance of epigenetic in obesity.
Response: We have added the following sentences at the end of third paragraph of the introduction.
“Epigenetic modifiers, including methyltransferases, impact metabolic disease states such as obesity and Type 2 diabetes [17]. Epigenetic regulation not only contributes to disease development, but is also relevant in responses to disease. Hence, understanding the role of the methyltransferases in adipocyte function is merited.”
[17] - (PMID: 30982733)
Reviewer 2 Comment: Line 39 you could be start with the characteristic and uses of 3T3-L1 cells. Why is important explore cellular mechanisms in these kinds of cells? is a model of diabetes or obesity and metabolic syndrome.
Response: We thank the reviewer for this comment. This information will help our study to be better understood by a broader audience. We added this statement to fourth paragraph of the Introduction:
“…. we investigated how G9a and GLP impact adipocyte function and gene expression in 3T3-L1 adipocytes. The 3T3-L1 adipocyte cell line is widely used to examine the basic cellular mechanisms associated with obesity and diabetes.”
Reviewer 2 Comment: Line 58 you mentioned “However, experiments using a G9a/GLP specific inhibitor that suppresses its catalytic function, BIX-01294, had no effect on IRα and phosphorylated AKT levels”, is important add in what kind of animal model, clinical study or what kind of cellular line found these results.
Response: Sorry for the oversight, we have added additional details on this reference. The revised text present in the introduction is shown below.
"Additional in vitro experiments in HepG2 cells have shown that loss of G9a protein levels results in decreased expression of insulin receptor alpha (IRα) and phosphorylated AKT and increased G9a expression could prevent fatty acid induced insulin resistance (3). However, experiments using a G9a/GLP specific inhibitor that suppresses its catalytic function, BIX-01294, had no effect on IRα and phosphorylated AKT levels in HepG2 cells (3)."
Reviewer 2 Comment: Line 77-79 this paragraph “Considering G9a and GLP expression in adipocytes contributes to metabolic health in mice, we designed our experiments to explore if loss of G9a and GLP could modulate the effect of tumour necrosis factor alpha (TNFα) action in adipocytes”, I suggest to change before of materials and methods, because is your principal aim.
Response: We are not clear what the reviewer means by this comment. Hence, we reworded these lines to the following, and hopefully that addresses the reviewer’s concern.
“Since in vivo studies indicate that G9a and GLP expression in adipocytes contributes to metabolic health in mice, we used an in vitro adipocyte model to determine if the loss of G9a and GLP expression could modulate the effect of tumor necrosis factor alpha (TNFα) action in adipocytes. TNFα is a pro-inflammatory cytokine produced and upregulated in adipose tissue macrophages in conditions of obesity and Type 2 diabetes [18], where it promotes lipolysis and induces insulin resistance.”
Reviewer 2 Comment: Line 91-104 I suggest al this paragraph is necessary rewriter because the writer it's a little confusing in the current version.
Response: We have rewritten lines 91-101 to hopefully eliminate any confusion. Thank you for the suggestion. This section now reads.
“Our studies are the first to demonstrate that loss of either G9a or GLP in mature adipocytes does not impact mono- or di-methylation of H3K9 and does not impact basal or TNFα-induced lipolysis. In addition, these two methyltransferases are present in a complex in the nucleus of adipocytes. Moreover, we observed that loss of both G9a and GLP in adipocytes did not have a significant effect on mono- or di-methylation of H3K9 suggesting that G9a and GLP are not the primary H3K9 methyltransferases in adipocytes under basal conditions. However, when G9a and GLP levels were depleted in TNFα-treated adipocytes, we observed a significant decrease in mono- and di-methylation of H3K9. Knockdown approaches revealed that loss of both G9a and GLP in adipocytes also resulted in an enhancement of TNFα-induced lipolysis and pro-inflammatory gene expression. Co-immunoprecipitation experiments revealed that p65, the major functional subunit of NF-kB, is only present in a complex with G9a and GLP under TNFα-stimulated conditions in adipocytes. Collectively, these novel studies provide insight into the cell autonomous roles of G9a and GLP in adipocyte gene expression and function that contribute to metabolic disease states.”
Reviewer 2 Comment: Line 107-114 You need to mention the plates you used and the number of cells you placed per well.
Response: We added that the cells were grown “to confluence”, and also added the statement: “All cell culture experiments were performed in Corning® Costar® tissue culture (TC)-treated 10 cm or multi-well plates.” Each specific methods section contains additional information about plates used or references that contain info about number of cells/well.
Our manuscript was initially flagged by the editor as being too similar to a previous publication of ours that focused on a different protein but utilized similar methods. The text duplications were mostly in the materials and methods section. To remedy the text duplications, we summarized and referenced the methods, and only included any important details that were different from the previous study. Based on this comment, we have gone through our material and methods section and added some details where applicable and not too duplicative to help the reader better understand the experiments performed in this study. We are confident that our experiments could be duplicated using the information contained within this manuscript and our referenced publications.
Reviewer 2 Comment: Line 142 could you add in the table 1 the gene accession or ID of gene bank basis in the primers that you used.
Response: We have updated Table 1 with this important information that will help our studies to be more reproducible.
Reviewer 2 Comment: Line 156-158 you need add a best description of the western blot, you did not mention which was the staining method and what equipment used.
Response: We added that “X-ray film” was used to detect the enhanced chemiluminescence signal: “Enhanced chemiluminescence signals produced by horseradish peroxidase-conjugated secondary antibodies were visualized on X-ray film [27].”
Reviewer 2 Comment: Line 222-233 you could place parentheses in each graph and add in the writing of figure 1, 2, 3,4, 5 and 6.
Response: We are not entirely clear what the reviewer means by this comment.
Reviewer 2 Comment: Line 338-348 you could place parentheses in each graph and add in the writing of figure 7.
Response: We are not entirely clear what the reviewer means by this comment.
Reviewer 2 Comment: Line 364-367 You could homogenize A. in the figure or A) in the writing. You could add a subsection or parenthesis to each graphic of the gene that was expressed and explain it in the writing of figure 8.
Response: We are not entirely clear what the reviewer means by this comment. We think our figures are presented in a concise manner with appropriate labeling.
Reviewer 2 Comment: Line 416-417 what is your opinion about these findings? I think is necessary write a supposed about this or add an image representing this hypothesis in 3T3-L1 cells.
Response: We are not entirely clear what the reviewer means by this comment. We indicated that interpretation of the data is that our studies strongly indicate that G9a or GLP alone are not the primary regulators of H3K9 methylation in cultured white adipocytes. Although this observation could be an artifact of the cell line we used, there is evidence from other investigators that mono- and di-methylation of H3K9 can be modulated by other methyltransferases including SETDB1 and SUV39H1 [40-42]. We do not think the addition of another image would enhance this interpretation.
Reviewer 2 Comment: Line 437-439 you have explored the role of PPAR alpha and its participation in adiponectin secretion in this kind of cells?
Response: We have not performed any studies on PPAR alpha. There is some evidence in the literature the PPARgamma agonists can promote adiponectin secretion, but we also did not examine PPARgamma in these studies.
Reviewer 2 Comment: Line 448-459 it is important to define the possible participation role of G9a and GLP, if it is protective or not, in the write of manuscript.
Response: We have indicated in other parts of the manuscript that the loss of G9a or GLP is associated with a metabolically unfavorable outcome. These data are summarized in paragraph 3 of the introduction. Our observations in this manuscript are consistent with interpretation that loss of G9a or GLP in adipocytes is associated with metabolically unfavorable conditions such as increased TNF-induced lipolysis and increased TNF-induced proinflammatory gene expression.
Reviewer 2 Comment: Line 475 the paragraph “under TNFα stimulated conditions” I think is better mentioned “under inflammatory stimuli conditions”.
Response: We revised the statement as follows: “Since we observed that p65 only associates with G9a and GLP under inflammatory conditions…..”
Reviewer 2 Comment: Line 513-516 I suggest all this information “Overall, our novel data showing that loss of both G9a and GLP enhances TNFα’s ability to induce pro-inflammatory gene expression and lipolysis are consistent with the metabolically unfavourable phenotype observed in G9a and GLP adipocyte-specific KO 515 mice”, you could be change in discussion section.
Response: We would prefer to keep this sentence in the conclusions as we don’t want to be too repetitive, and this is a primary conclusion of the paper. Thank you for considering this request.
Reviewer 2 Comment: Line 516-524 in this section of the conclusions you could be more concise in your writing of findings.
Response: We have reworded this section to be more concise. This section now reads:
“Our data indicate that G9a and GLP have a protective role in modulating the transcriptional activity of p65 in response to TNFα exposure. Although G9a and GLP expression affects the ability of TNFα to regulate gene expression and lipolysis, further studies are needed to determine if the observed effects are dependent on G9a and GLP’s methylation of H3K9 and/or a non-histone substrate. Future studies can determine if these observations are specific to TNFα or if these pathways are similarly regulated by other pro-inflammatory cytokines when the expression of these methyltransferases is reduced. Overall, these studies underscore the complexity of epigenetic contributions to gene expression.”
Reviewer 2 Comment: Line 516-517 The principal role of G9a and GLP in adipocytes is protective?
Response: Our observations in this manuscript are consistent with interpretation that loss of G9a or GLP in adipocytes is associated with metabolically unfavorable conditions such as increased TNFα-induced lipolysis and increased TNFα-induced proinflammatory gene expression. So, this suggest that G9a and GLP may be protective in these conditions. We revised this statement in the conclusion as follows:
“Our data indicate that G9a and GLP have a protective role in modulating the transcriptional activity of p65 in response to TNFα exposure.”

Reviewer 3 Report
Methodology should be well described with details provided to enable reproducibility for example Section 2.8, how was protein expression quantified? Also, Section 2.7 talks about incubation medium without stating what exactly this medium is and its constituents. This goes for the entire methodology section.
Author Response
We appreciate the feedback from the three reviewers on our submitted manuscript “TNFα effects on adipocytes are influenced by the presence of lysine methyltransferases, G9a (EHMT2) and GLP (EHMT1)”. We are thankful that it was indicated that “this is an interesting study, and the experiments are carefully performed and clearly presented. We have addressed all the reviewers’ comments, but do not have the ability to perform additional experiments. Also, we are trying to meet the April 6 deadline from the Editor. Thank you for your consideration of our revised manuscript to Biology (Manuscript ID: biology-2289197).
In addition to the response below, we have attached a marked version of our revised manuscript.
Reviewer 3 Comment: Methodology should be well described with details provided to enable reproducibility for example Section 2.8, how was protein expression quantified? Also, Section 2.7 talks about incubation medium without stating what exactly this medium is and its constituents. This goes for the entire methodology section.
Response: Our manuscript was initially flagged by the editor as being too similar to a previous publication of ours that focused on a different protein but utilized similar methods. The text duplications were mostly in the materials and methods section. To remedy the text duplications, we summarized and referenced the methods, and only included any important details that were different from the previous study. Based on this comment, we have gone through our material and methods section and added some details where applicable and not too duplicative to help the reader better understand the experiments performed in this study. We are confident that our experiments could be duplicated using the information contained within this manuscript and our referenced publications.
Round 2
Reviewer 1 Report
The authors's have been responsive to the comments of all 3 reviewers. The manuscript is improved and the study is interesting and provides new information.
Author Response
Thank you for finding study interesting and novel and our manuscript improved. We appreciate your feedback.
Reviewer 2 Report
Line 86 please to change “in vitro” by “in vitro”
Line 207 to change “P” by “p” because this P is incorrect.
Author Response
Thank you for your additional edits! We have made the recommended changes. Please see attached revised manuscript.
